# Multimodal Remote Home Monitoring of Lung Transplant Recipients during COVID-19 Vaccinations: Usability Pilot Study of the COVIDA Desk Incorporating Wearable Devices

**DOI:** 10.3390/medicina59030617

**Published:** 2023-03-20

**Authors:** Macé M. Schuurmans, Michal Muszynski, Xiang Li, Ričards Marcinkevičs, Lukas Zimmerli, Diego Monserrat Lopez, Bruno Michel, Jonas Weiss, René Hage, Maurice Roeder, Julia E. Vogt, Thomas Brunschwiler

**Affiliations:** 1Division of Pulmonology, University Hospital Zurich, 8091 Zurich, Switzerland; 2Faculty of Medicine, University of Zurich, 8032 Zurich, Switzerland; 3IBM Research Europe, 8803 Rüschlikon, Switzerland; 4Department of Mathematics, ETH Zurich, 8092 Zurich, Switzerland; 5Department of Computer Science, ETH Zurich, 8092 Zurich, Switzerland; 6Department of Biosystems Science and Engineering, ETH Zurich, 4058 Basel, Switzerland

**Keywords:** home monitoring, COVID-19 vaccination, lung transplant, digital health, respiratory disease, disease management, chronic disease, patient monitoring

## Abstract

*Background and Objectives*: Remote patient monitoring (RPM) of vital signs and symptoms for lung transplant recipients (LTRs) has become increasingly relevant in many situations. Nevertheless, RPM research integrating multisensory home monitoring in LTRs is scarce. We developed a novel multisensory home monitoring device and tested it in the context of COVID-19 vaccinations. We hypothesize that multisensory RPM and smartphone-based questionnaire feedback on signs and symptoms will be well accepted among LTRs. To assess the usability and acceptability of a remote monitoring system consisting of wearable devices, including home spirometry and a smartphone-based questionnaire application for symptom and vital sign monitoring using wearable devices, during the first and second SARS-CoV-2 vaccination. *Materials and Methods*: Observational usability pilot study for six weeks of home monitoring with the COVIDA Desk for LTRs. During the first week after the vaccination, intensive monitoring was performed by recording data on physical activity, spirometry, temperature, pulse oximetry and self-reported symptoms, signs and additional measurements. During the subsequent days, the number of monitoring assessments was reduced. LTRs reported on their perceptions of the usability of the monitoring device through a purpose-designed questionnaire. *Results*: Ten LTRs planning to receive the first COVID-19 vaccinations were recruited. For the intensive monitoring study phase, LTRs recorded symptoms, signs and additional measurements. The most frequent adverse events reported were local pain, fatigue, sleep disturbance and headache. The duration of these symptoms was 5–8 days post-vaccination. Adherence to the main monitoring devices was high. LTRs rated usability as high. The majority were willing to continue monitoring. *Conclusions*: The COVIDA Desk showed favorable technical performance and was well accepted by the LTRs during the vaccination phase of the pandemic. The feasibility of the RPM system deployment was proven by the rapid recruitment uptake, technical performance (i.e., low number of errors), favorable user experience questionnaires and detailed individual user feedback.

## 1. Introduction

Lung transplantation is the ultimate treatment option for patients with end-stage lung disease. Lung transplant recipients (LTRs) need a long-term, close follow up by dedicated specialists to provide continuous medical care for their complicated medical condition [1]. Digital health plays a critical role in the response of healthcare organizations during the COVID-19 pandemic, especially in vulnerable populations, including solid organ transplant recipients. While telemedicine offers a real-time patient–provider encounter, the inability to obtain vital signs during virtual visits is a potential limitation. Remote patient monitoring (RPM), sometimes also referred to as telemonitoring [1], uses portable or wearable devices in patients’ homes to collect and digitally transmit physiological data and questionnaire information as well as self-recorded data via an application interface to clinicians. For example, sensor data from a wearable device is transmitted by Bluetooth to the smartphone. The data is then transferred via the Global System for Mobile Communication (GSM) to the cloud to be stored and processed. It is then made available to the physician through a web interface either in real time or slightly delayed due to data processing. RPM can be used for regular patient care to monitor remotely, improve patient adherence to treatment or intervene early in case of detected complications. It may contribute to reducing the number of personal consultations with the physician. It can also be used in special situations such as virus outbreaks/pandemics or to monitor patients during or after specific interventions, for example, to document vaccine responses [2,3,4]. The use of RPM in LTRs is primarily motivated by the fact that it has the potential to improve patient outcomes, including better and earlier diagnosis of complications, increase quality of life by facilitating transparency and reassurance of the current patient condition and reducing the personal consultation frequency [1,2,3]. Some telemedical remote monitoring programs do not rely on measuring devices but focus only on information exchange based on software applications using mobile devices or regular computers in solid organ transplantation recipients [5]. Only a small number of randomized controlled trials have investigated mobile health interventions in LTRs focusing on patient education, spirometry, self-management and a computer-based decision system for triage [6,7,8,9].

The use of wearable devices to detect COVID-19 at an early stage is increasingly being studied in various populations, including LTRs [10,11]. The use of wearable devices for monitoring the vaccine reactions of the COVID-19 vaccines has been studied in the general population, but data from vulnerable populations, for example, LTRs, has not been analyzed [12]. An additional use could be to monitor patients in the hospital (on the regular ward) or at home when their condition is not entirely clarified. More frequent monitoring is advisable to detect possible deterioration or a trend for improvement, for example, for vulnerable patients with COVID-19 [11].

LTRs are familiar with home monitoring by home spirometry, body temperature and blood pressure measurements, self-observation and documentation of symptoms, such as respiratory symptoms, liquid intake, stool frequency, stool consistency, and blood sugar measurements [13]. Any additional aspects of self-monitoring are tailored to the individual comorbid conditions or the current disease state, such as heart or renal failure, history of recurrent impaired intestinal passage or obstipation. As part of the pre-transplant and post-transplant education process, the patients are taught to closely monitor their health status (i.e., signs, symptoms and self-measurement results). However, the recorded data is currently not made immediately available to the physician but is used for self-management and to prompt patients to contact the transplant center for advice. LTRs are instructed to contact the transplant pulmonologist by telephone if thresholds for signs or symptoms are reached. Many of these monitoring results are documented manually by the patients and are also discussed routinely in outpatient visits with the physician in the post-transplant clinics [13]. This monitoring is part of the standard procedures after lung transplantation. It is considered an integral part of the follow up led by the notion that early diagnosis and specific treatment of potentially serious clinical complications can reduce associated morbidity and mortality [13,14]. Despite self-monitoring being practiced widely and consistently and the provision of clear instructions concerning when to contact the health care provider, there is perceived reduced reliability in reporting new or worsening symptoms since the patient determines autonomously when to contact their health care professional [14]. Optimization of RPM for this population may help to recognize situations where early remote interventions may potentially be beneficial for managing health complications early.

As a result of the pandemic, LTRs were often reluctant to come to personal outpatient visits in the hospital due to concerns about being infected with SARS-CoV-2, either during the travel to the hospital or at the hospital itself. Telehealth consultations were the obvious temporary solution for many aspects that could be discussed on a telephone or a video call. We also noted that patients were more reluctant to call in due to observed problematic symptoms, signs or measurements for the same reason: not wanting to appear in person at the hospital due to perceived risks. This prompted us to consider remote monitoring of the patients’ health conditions, which included measurements by wearable devices (accelerometer, finger pulse oximetry, continuous temperature measurements and self-measurement results from spirometry, single temperature measurements, blood pressure readouts and signs and symptoms observed by the patients such as stool frequency, weight alterations, etc.).

To achieve this in a systematic way, we planned to assess the feasibility and acceptability of such a monitoring system using an adapted set of measurement devices assembled as the COVIDA desk. It is a derivative version of the CAir desk dedicated to monitoring chronic obstructive pulmonary disease (COPD) patients [15].

The main goal of this pilot study was to use and evaluate the COVIDA desk in the context of assessing the symptom burden in stable LTRs receiving their first two vaccination doses of the COVID-19 vaccine. As this was an observational study, the patients were instructed to contact the transplant pulmonologist according to their usual rules of self-monitoring. The focus was on the feasibility and utility of the COVIDA desk monitoring tool and its acceptance among the pilot population of LTRs.

## 2. Methods

### 2.1. Study Design

We conducted an observational usability pilot study. LTRs underwent 6 weeks of disease home monitoring with the COVIDA desk during the period of the two SARS-CoV-2 vaccinations without interventions or modifications to their established treatment regimen. The participants were instructed to perform daily measurements in accordance with the schedule depicted in Figure 1. The monitoring was divided into an intensive and a reduced monitoring phase for each of the received vaccinations. The intensive phase was scheduled on the day of the first vaccination and two days before the second vaccination. The duration of the intensive and the reduced phase was 8 days and approximately 18 days (depending on the interval between the first and second vaccination), respectively.

After the study period, participants reported their perceptions of the usability of the COVIDA desk with a purpose-designed questionnaire, which included the User Experience Questionnaire questions (UEQ) and additional questions relating to the different monitoring devices to detect any adverse device effects and asses the acceptance among the LTRs [16,17]. No specific clinical activity was triggered by the COVIDA desk RPM system, i.e., no alerts were sent, even for abnormal readings. However, patients could observe measurement results on their smartphone apps and as usual, call their transplant pulmonologist if there was any uncertainty.

The study team was able to track the adherence of patients to the study protocol during the trial. It was not part of the protocol to encourage the LTRs to use the COVIDA desk devices more often in the case of low adherence. However, in the case of two consecutive days without data upload, patients were contacted to resolve any technical or usability issues.

This study did not fall within the scope of the Human Research Act (HRA) and did not require formal authorization from the ethics committee. The Ethics Committee of the Canton of Zurich confirmed this in the BASEC Request (2021-00480).

### 2.2. Study Patients

For this study, we applied convenience sampling. Ten participants with stable conditions after lung transplantation attending the lung transplant outpatient clinic at the Division of Pulmonology, University Hospital Zurich, Switzerland, consented to participate in the study after receiving information on the study procedures.

### 2.3. Study Materials

The COVIDA desk (Figure 2) is a novel, custom-built disease home monitoring system. It combines multiple sensors in a compact format with a single power plug for device charging. All components of the COVIDA desk are Conformité Européenne (CE)-certified.

Physical activity (i.e., step count), sleep (i.e., duration and awakenings) and heart rate were measured using a multisensory wearable wrist-worn device (Inspire 2, Fitbit Inc., San Fransico, CA, USA, Figure 2, No. 1) containing a triaxial accelerometer and a photoplasmogram. The device was worn during daily and nightly activities and had to be recharged for an hour after a few days of recording. The data is transferred via Bluetooth to the smartphone in the case of the proximity of the sensor to the COVIDA desk.

The continuous pulse oximetry was a voluntary measurement since it was not considered the primary outcome variable for vaccine-related adverse events. We encouraged the patients to use the finger-ring-shaped sensor (i.e., O2ing, Wellue, Diamond Bar, CA, USA) Figure 2, item No. 2) during the intense monitoring phase to observe oxygen saturation and oxygen variations during sleep (i.e., nocturnal measurement) or the day (i.e., physical activity/exercise). The measured data was transferred to the smartphone by Bluetooth in the morning, after the measurement phase.

A handheld infrared thermometer device (i.e., FTN Infrarot, Medisana, Neuss, Germany, shown in Figure 2, No. 3) was used daily to measure the forehead temperature. The measurement had to be transcribed manually into the smartphone questionnaire.

The core body temperature (CBT) was considered a main monitoring parameter, as increased CBT has been reported after vaccination in other studies [18]. Therefore, we considered this measurement compulsory and encouraged the patients to wear the CORE device (i.e., CORE, greenTEG AG, Ruemlang, Switzerland, shown in Figure 2, No. 4) every day, except when they were taking a shower or a bath. During these times, the device should be reloaded with the provided cable and the data should be transferred via Bluetooth to the CORE app on the smartphone of the COVIDA-desk.

Daily spirometry recordings were obtained with a portable spirometry device (i.e., Air Next Spirometer, NuvoAir, Stockholm, Sweden, shown in Figure 2, No. 5), which was connected to the smartphone via Bluetooth to perform and display test results. The values obtained were forced expiratory volume in 1 s (FEV_1_) and forced vital capacity (FVC). All LTRs are trained early after lung transplantation to perform home spirometry and to produce reproducible maneuvers that comply with published guidelines. Therefore, all LTRs had already performed home spirometry on a regular basis before participation in this study [13].

The smartphone (i.e., Galaxy A320, 2017, Samsung Group, Seoul, Republic of Korea, shown in Figure 2, No. 6) contained purpose-designed apps for user interaction and data visualization. All sensors are accessible via the smartphone through a Bluetooth connection. Further, all sensor data are transferred to the cloud storage through the smartphone by the GSM network. Data transfer and data storage on the cloud were performed with encryption, as required by data protection regulations for sensitive personal information. Further details about technical, cloud, and backend solutions can be found in the work of Gross et al. [19].

Symptoms and potential adverse events from the vaccination were assessed by a daily questionnaire with predefined answers of intensity and one open question, all provided on the smartphone display that patients were prompted to answer daily from 6:00 to 23:00. The smartphone displayed push notifications when the daily questionnaire was incomplete. Participants answered the questions directly in the questionnaire (i.e., COVIDA App).

To assess device usage and identify days of adherence, we defined thresholds for each sensor. An overview of these thresholds for each sensor, alongside the details on recording modalities, is provided in Table 1.

To assess the association between the age and time since lung transplantation with adherence, a single adherence statistic for each patient was calculated across all modalities and compared statistically across the patients below and above the median age and across those below and above the median time since transplant.

### 2.4. Statistical Analysis

One of the goals of our study is to investigate the physiological and behavioral responses of vulnerable populations, i.e., solid organ transplant recipients, to COVID-19 vaccines. Our remote patient monitoring setup consisted of portable devices or wearables, allowing us to collect physiological data and questionnaire information. We aim to explore changes in physiological data, the content of the questionnaire and self-report data acquired from the LTRs; therefore, we carried out statistical analyses to determine the most frequent and intensive symptoms, signs, and adverse events up to 4 post-vaccination days (i.e., from the 1st to the 4th day after receiving the first or second dose of COVID-19 vaccines). We first ran the Wilcoxon signed-rank tests at 0.05 significance level to compare medians of each biomarker between two periods, i.e., from the 1st to the 4th day of the intervention period after receiving the first and second vaccine dose vs. from the 5th to the 8th day of the control period after receiving the first and second vaccine dose, then calculated the effect size (*r*) for the Wilcoxon signed-rank tests. The two-sided Wilcoxon rank sum test was used to compare adherence across the age categories and time-since-transplant categories. We interpret an outcome as statistically significant if the *p*-value is smaller than 0.05. For the effect size (*r*), values around 0.1, 0.2 and 0.5 can be interpreted as the effect of small, medium and large magnitude, respectively. All results are presented using descriptive statistics in the format “median (range)” unless stated otherwise.

## 3. Results

Ten LTRs were recruited before a planned SARS-CoV-2/COVID-19 vaccination. Detailed summaries of the characteristics of the patient cohorts are reported in Table 2. All participants completed the predetermined study period and did not experience any adverse events resulting from the monitoring components used. One patient did not receive the second vaccination due to a COVID-19 diagnosis after the first vaccination; therefore, for this subject, only the data from the first vaccination were included.

### 3.1. Patient Adherence and Technical Considerations

Adherence was assessed for both wearing and downloading data from the wearables as well as providing answers to the questionnaire items, which included some self-measurement results obtained by the participant. Adherence results are presented in Figure 3 and Figure 4. In general, adherence to the recommended (mandatory) monitoring features during the intensive phase was very high, and adherence was lower for voluntary measurements. Adherence to monitoring was hardly ever affected by technical complications since no major technical issues were noted, and the few minor technical issues were quickly resolved. No significant associations were detectable between patient age and adherence or between time since transplant and adherence.

Most symptoms and signs of health and disease were transferred via the COVIDA desk GSM connection to the study team daily. The symptom frequency and e-health monitoring parameters are summarized in Figure 5. The most frequent symptoms after two sets of vaccinations were Fatigue (87.5%), local pain at the injection site (81.3%), sleep disturbance (68.8%), followed by headache (43.8%), Hypoglycemia (42.9%) and injection site swelling (31.3%). Figure 6 shows the symptom frequency during the first eight days after the vaccinations for both vaccination periods, with disturbed sleep, local pain and fatigue being observed most frequently. Figure 7 depicts the duration of each symptom for individual patients with sleep disturbance and fatigue lasting the longest, up to approximately one week.

Symptom frequency is indicated by color intensity, whereby darker colors represent higher frequency. The size of the square indicates the number of vaccinated patients affected by respective symptoms. Abbreviations: Vomit, vomiting; Fatigue, tired; Smell, change in smelling or tasting capability; Sleep-Disturb, change in sleep quality; Less-Stools, reduced stool count as compared to usual stool frequency; Local-Swelling, swelling of the injection site; Local-Erythema, erythema at the injection site; Local-Pain, pain at the injection site; Stools-Harder, change in stool consistency (harder); Dyspnea, shortness of breath.

The COVIDA desk was evaluated as an entire monitoring tool by the LTRs in the used user experience questionnaire (UEQ). The results are shown in Figure 8. In addition, the main components were evaluated separately to identify preferences or problematic aspects.

### 3.2. Patient Satisfaction and Acceptance for Continued Use

Regarding the COVIDA desk, in general, six (60%) patients indicated that they would be willing to use the device further in its present form. Four (40%) patients indicated that they would not want to continue the use. Furthermore, various comments were made about the device, apps and difficulties encountered while using the hardware and software (see Table 3).

Detailed feedback on the different components of the desk is provided in Table 4. The willingness to continue using the whole COVIDA desk or the components of it was answered affirmatively by 60% of patients for the COVIDA desk and by 10, 40, 70, 80 and 100% of LTRs for the ring-oximeter, CORE temperature device, wrist-accelerometer, infrared thermometer, COVIDA application and spirometer, respectively (see Table 4).

### 3.3. Physiological Data and Effect Size Estimation

The analysis of the physiological data obtained from various sensors, such as temperature measurements, oxygen saturation, heart rate and temperature monitoring, spirometry and physical activity data based on measurements of the wrist-worn accelerometer, was performed by comparing the immediate post-vaccination phase (days 1–4) to the reference period on day 5–8 after the vaccination (assuming that these measurement parameters would have returned to “normal” or a steady state by this time). None of these measurements in the immediate post-vaccination period differed significantly from the reference phase. However, using a one-sided statistical test based on a directed hypothesis in selected parameters, certain trends were detectable (see Figure 9 and Figure 10).

Figure 9 shows the values of the effect size (*r*) for symptom frequency-based biomarkers, including headache, fatigue, and sleep-disturbed biomarkers, calculated between the intervention and control periods for both post-vaccination periods, i.e., after the first and second vaccinations. We find two large positive effects for the headache and fatigue biomarkers. In particular, the patients had more headaches and felt more tired during the intervention period vs. the control period for both vaccinations. Additionally, we detect a moderate positive effect for the sleep-disturb biomarker.

Figure 10 shows the values of the effect size (*r*) for symptom intensity-based biomarkers, such as stool-count, min. blood sugar, max. blood sugar, and CBT biomarkers calculated between the intervention and control periods for both post-vaccination periods (i.e., after the first or second vaccination). The magnitude and direction of the effect sizes supported by our statistical comparison reveal the high discriminability of stool-count only. This finding suggests that lung transplant recipients suffered from reduced stool frequency during the intervention period after the first and second vaccination when compared to the control period.

## 4. Discussion

We report on the experiences with the first clinical application of the COVIDA desk, a remote home monitoring device with multiple measurement features, including spirometry, temperature, activity tracker and questionnaires. In this usability study, the COVIDA desk performed well in technical terms. It was well accepted by the participants with a stable condition after lung transplantation receiving the first two SARS-CoV-2 vaccinations. The general adherence to the measurement schedule was considerably high. We found no major barriers or obstacles regarding the usage of the technical equipment or software. Usability ratings were high based on the user experience questionnaire evaluation. More than half of the LTRs indicated that they would be willing to continue using the COVIDA desk. The spirometer, the infrared thermometer, the wrist-worn accelerometer and the COVIDA app questionnaire feature were the components with the best acceptability for further use among LTRs. Some suggestions were made by patients on how the monitoring system could be improved.

Although the study of the clinical responses to the vaccinations was not the main objective of this pilot study, it, nevertheless, is interesting to take note of the observed symptoms: The main symptoms observed after the vaccinations were local pain at the injection site, fatigue, sleep disturbances and headache of which fatigue and sleep disturbances lasted longer than has previously been reported. A reduced stool count following COVID-19 vaccination was a relevant additionally observed symptom by the LTRs, of which 50% had cystic fibrosis as an underlying condition.

The number of studies investigating telemedicine equipment has increased considerably in recent years. Nevertheless, there are few studies that are already available and reporting on experiences in the context of COVID-19 in solid organ transplant recipients, in particular for lung transplant recipients [1,2,3,11,12,20,21]. The potential advantages of telemedicine, particularly telemonitoring with remote monitoring systems, are increasingly being recognized and investigated. Usability and adherence of such RMS are closely related, but both aspects are rarely explored in detail [15,21].

Patient adherence to the use of the COVIDA desk was moderately high when considering all components of the monitoring system but consistently high when considering specific devices, namely, the wrist-worn accelerometer, the infrared thermometer and the spirometer (shown in Figure 4). Except for core body temperature and pulse oximetry on day one after vaccination, adherence varied between 40% and 100% in our pilot population. Kohlbrenner reported on individual adherence rates of 25–80% for symptom-burden questionnaires and spirometry among asthma and COPD patients using a similar RMS over four weeks [15]. Other studies have reported adherence rates to spirometry with mobile health devices of 59–97% [7,9]. Our LTRs were used to performing spirometry before this study, which may have contributed to the higher adherence rate for spirometry measurements. Considering the wide range of age among the study population and the large variability in the time since transplant, one may suspect an influence of these variables on adherence to monitoring. We did not detect a statistically significant association between patient age and adherence and not for time since transplant and adherence in this small pilot population.

We were interested in assessing the user experience and acceptability of the COVIDA desk and its specific components. From a general point of view, the monitoring system was rated highly as evaluated by the user experience questionnaire: The adjectives and rating chosen to describe the COVIDA desks were favorable for all qualities, as depicted in Figure 6 by predominately positive responses marked in green.

Future evaluations of the COVIDA desk usage should consider using the remote monitoring satisfaction survey developed by Finkelstein et al., as it was designed explicitly for LTRs [22]. Other questionnaires may also be considered in addition to the UEQ used here, which we considered the most appropriate for a pilot trial [16,17,23].

Sixty percent of the users would be prepared to continue using the monitoring system, and 80–100% would continue using the spirometer, accelerometer and the smartphone-based questionnaire, showing that specific devices are more acceptable than others. Acceptability was likely influenced by some user experiences that were mentioned critically and may serve to improve some of the devices. For example, the falling off of the core thermometer was noted by several patients (see Table 3). In general, the COVIDA desk was rated highly in the user experience questionnaire and compared well to the benchmark population [24].

Considering symptom burden or adverse event frequency and duration of these symptoms after vaccination, our pilot study provided some noteworthy findings: The most frequent adverse events noted by the LTRs were fatigue (87%), local pain at the injection site (81%), disturbed sleep (68%), headache (44%), hypoglycemia (43%) and local swelling at the injection site, which were very similar to the findings of Hallett et al. [25] observed in heart and lung transplant recipients and Ou et al. [26] in SOT recipients in general. In addition to fatigue, we noted significant sleep disturbances and hypoglycemia among our pilot population. These aspects were not explicitly assessed in the study conducted by Hallett et al. [25] but have partly been observed by others [27].

The symptom burden after COVID-19 vaccinations among our LTRs appeared to be higher than reported for the vaccinations of the general population and some vulnerable populations as well: The proportion of LTRs experiencing any kind of adverse event was high in comparison to the proportion experiencing such symptoms in the general population, where generally half the participants did not report any local or systemic reaction following the vaccination [12].

The duration of the adverse events was longest for sleep disturbance (i.e., 6–8 days) and fatigue (i.e., 6–8 days), followed by pain at the injection site (i.e., five days) (see Figure 5). The duration of the adverse events is generally 5–7 days for the mRNA vaccines, based on the initial study results. Gepner et al. [12] reported a substantial decline in adverse events on day three post-vaccination in a non-transplant setting, meaning that less than 20% of the participants experienced fatigue, headache or muscle pain. Although several safety assessments in solid organ transplant recipients have described the type of adverse events observed and sometimes the severity, there is hardly any published data on the duration of these symptoms. Thus, the total symptom burden is largely not known for this population [18].

In contrast to most other studies, we specifically queried patients about stool frequency and consistency as well as blood sugar levels during the post-vaccination phase since we had anecdotally noticed variations in these parameters during the first months of vaccine roll-out [28]. No clearly significant findings or trends were observed in this pilot population regarding these parameters. However, some LTRs appeared to have decreased stool frequency and increased stool consistency, which was only detectable when comparing these parameters in the post-vaccination phase with a reference phase considered baseline, as explained below.

In addition to patient-reported outcomes, such as adverse events and symptoms experienced following the COVID-19 vaccinations, we gathered data on physiological changes as assessed by various sensors, such as temperature measurements, oxygen saturation, heart rate and temperature monitoring, spirometry and physical activity data based on measurements of the wrist-worn accelerometer.

The analysis of this data was compared to a reference period on days 5–8 after the vaccination, assuming that these measurement parameters will have returned to a “normal” or steady state by this time [12]. None of these comparisons (i.e., between the immediate postvaccination phase day 1–4) generated significantly different measurement results from the reference phase (i.e., days 5–8) using two-sided significance tests. This may have been expected since the sample size was small. However, since we had defined a directed hypothesis for selected parameters before the study, we performed a one-sided significance test. We detected evidence for trends in a few comparisons, namely reduced stool frequency and increased stool consistency during the first week post-vaccination. Of course, these findings must be treated with caution and merely serve to generate a hypothesis for further evaluations.

Gepner et al. did show that both symptomatic and asymptomatic participants had substantial objective physiological changes during the first days after vaccination, regardless of their subjective reports underscoring the importance of obtaining physiological data in addition to self-reported questionnaire information when performing clinical trials [12]. Whereas self-reported trends are widely described in the scientific literature, no study or vaccine clinical trial for LTRs has reported the comprehensive effects of the COVID-19 vaccine on physiological measures [12].

The results obtained from the adverse event and physiological monitoring during the first and second COVID-19 vaccination were mainly in line with published information from other vulnerable patient populations [14]. However, our pilot study population had a greater symptom load and some longer-lasting adverse events compared to previous reports [18].

Our study has several limitations. First, our pilot study population is small and may not adequately represent the vaccinated LTR population in Switzerland or elsewhere. It was designed as a feasibility study. Therefore, the small sample size is plausible. Nevertheless, the proportion of those who reported local and systemic reactions and the type of reactions noted were similar to those observed in larger studies [29,30], particularly those using a similar methodology [12]. The small sample size also limits the power in testing for associations between patient adherence and characteristics, such as age or time since transplantation. Second, most patients received the BNT162b2 vaccine (by Pfizer-Biontech), which was more widely available in Switzerland at the time. Our findings are, therefore, mainly representative of the BNT162b2 vaccine rather than the underrepresented Moderna vaccine.

Given the similarities in the local and systemic reactions observed between different COVID-19 mRNA vaccines, we believe that the choice of the vaccine was not relevant to the main objective of this feasibility study.

## 5. Conclusions

We showed both the feasibility and acceptance of the COVIDA desk at a patient level by high adherence to the monitoring scheme and by providing favorable user feedback. We also demonstrated the technical feasibility of daily monitoring with a multisensory system and reliable GSM-based data transmission of a large portion of the high-resolution data obtained. Adherence levels to monitoring procedures were high, and more than half of the patients would support the continued use of the COVIDA desk. The symptom load in stable LTRs receiving baseline vaccinations for COVID-19 appears to be larger and lasts longer for some symptoms, such as fatigue and sleep disturbances, than previously reported. The information obtained in this pilot is being used to improve some features of the COVIDA desk for use in a larger study investigating more meaningful outcomes, such as complication rates (graft function, rejection episodes, infection, rehospitalization, mortality) and frequency of remote and physician contact visits as well as quality of life.

## Figures and Tables

**Figure 1 medicina-59-00617-f001:**
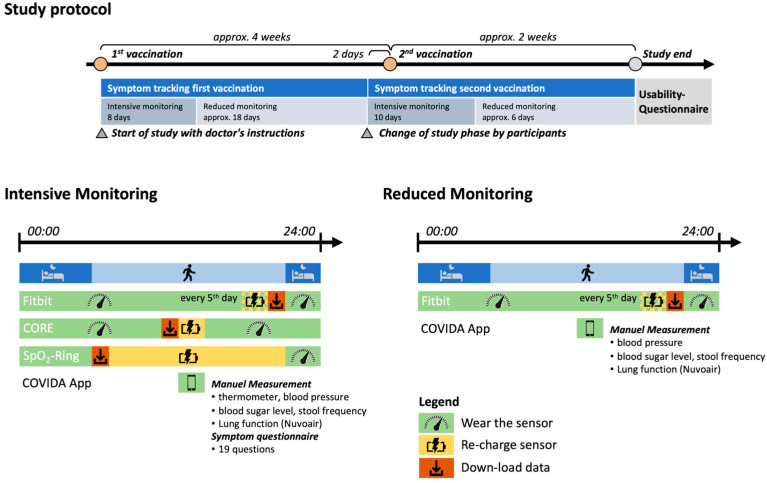
Daily measurement schedule during the intensive monitoring and reduced monitoring study periods. This figure was used to explain the study procedures to the patients and handed out as instructions for later reference (translated version since original was in German).

**Figure 2 medicina-59-00617-f002:**
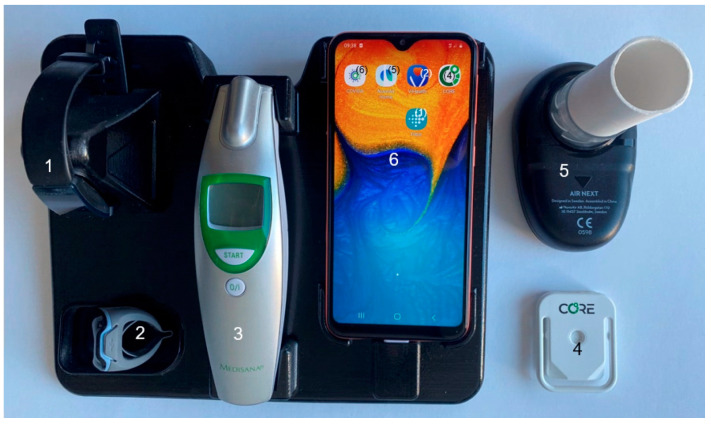
The COVIDA desk setup for the usability study contains a collection of sensors and monitoring modalities. Items in the picture are numbered and explained: (1) Wrist-worn accelerometer and heart rate sensor on magnetic charging interface; (2) finger ring pulse oximeter for continuous measurement of heart rate and oxygenation saturation; (3) infrared thermometer for forehead temperature measurement; (4) continuous core body temperature measurement device; (5) spirometer with mouthpiece for measurement of lung function; and (6) smartphone with daily questionnaire in the COVIDA app. The respective applications on the smartphone screen are indicated in brackets.

**Figure 3 medicina-59-00617-f003:**
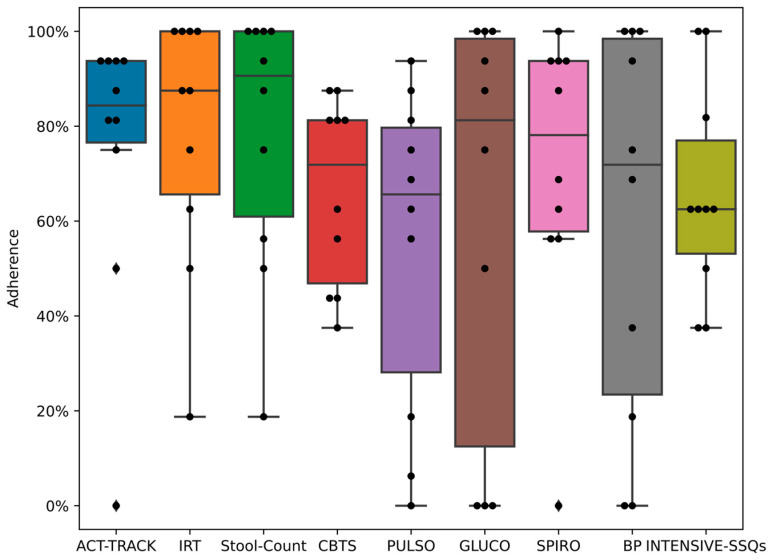
Adherence to monitoring modalities during intensive monitoring period for both vaccinations combined. The x-axis shows the different modalities studied while the y-axis shows box plots of adherence. PULSO: PULSOoximeter; CBTS: Core Body Temperature Sensor; ACT-TRACK: ACTivity-TRACKer; IRT: InfraRed Thermometer; INTENSIVE-SSQs: Intensive Symptom and Sign Questionnaires; SPIRO: SPIROmeter; BP: Blood Pressure; GLUC: GLUCOmeter.

**Figure 4 medicina-59-00617-f004:**
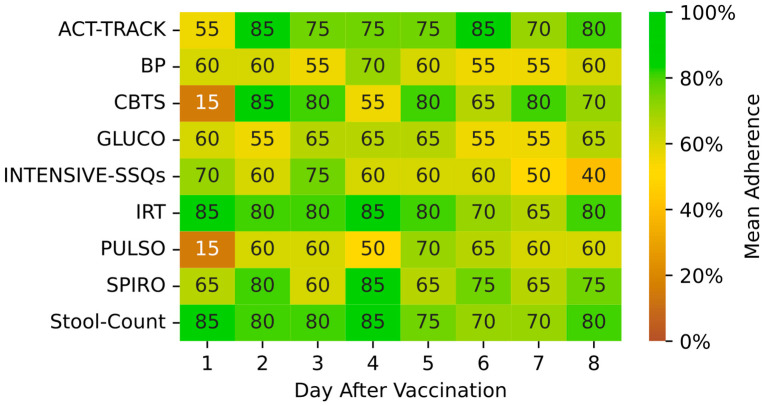
Adherence to monitoring modalities heatmap for both monitoring periods combined (the 1st and 2nd vaccination) for the whole study population. Green-color fields indicate very high adherence, yellow fields indicate moderate adherence and brown fields indicate low adherence. On the x-axis, the days after vaccination are presented for intensive monitoring phase, i.e., the first 8 days, while the y-axis corresponds to modalities. Adherence to INTENSIVE-SSQs means that all questions were answered. Abbreviations: PULSO: PULSOoximeter; CBTS: Core Body Temperature Sensor; ACT-TRACK: ACTivity-TRACKer; IRT: InfraRed Thermometer; INTENSIVE-SSQs: Intensive Symptom and Sign Questionnaires; SPIRO: SPIROmeter; BP: Blood Pressure; GLUC: GLUCOmeter.

**Figure 5 medicina-59-00617-f005:**
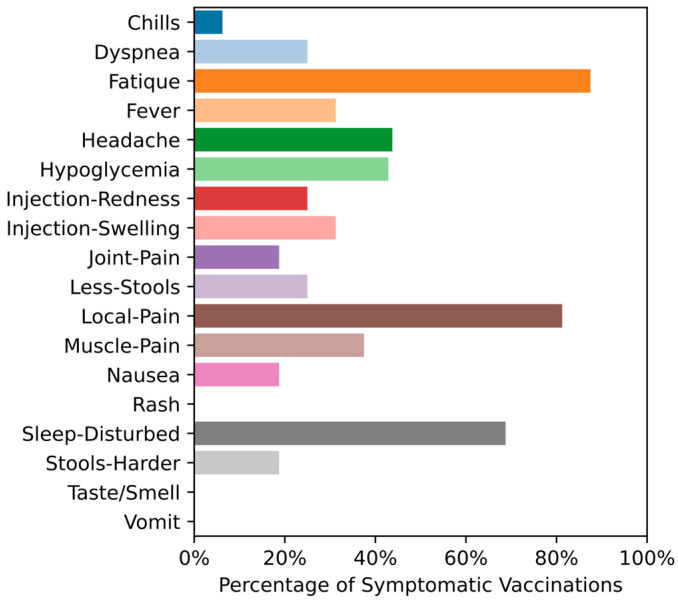
Symptom frequency purely based on whether the patients had the specific symptom during the two vaccination phases.

**Figure 6 medicina-59-00617-f006:**
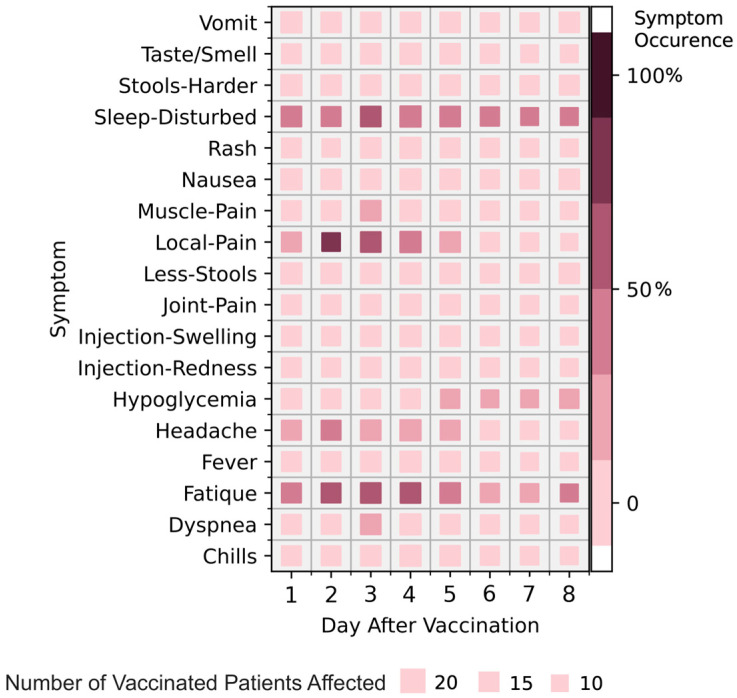
Symptom frequency for both monitoring periods of the 1st and 2nd vaccination combined. Symptom frequency for both monitoring periods is shown.

**Figure 7 medicina-59-00617-f007:**
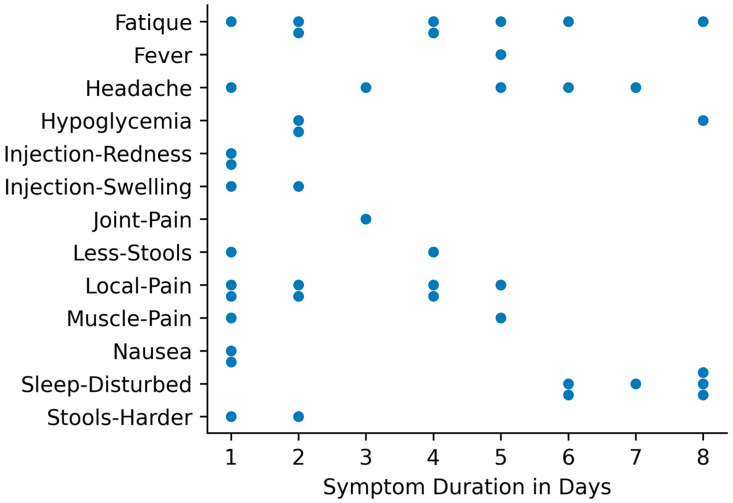
The duration of symptoms documented during the intensive monitoring phase is shown. Each data point represents a patient with the respective symptom indicating the duration of the symptom.

**Figure 8 medicina-59-00617-f008:**
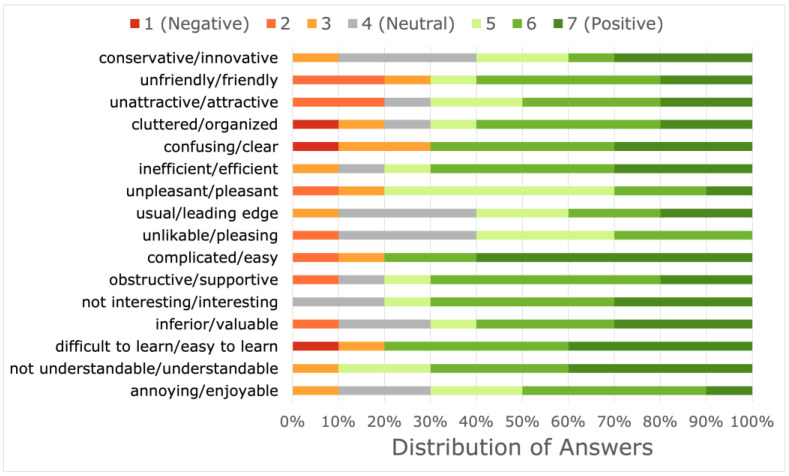
User experience questionnaire evaluation for selected questions. Red and orange colors indicate negative assessments, gray indicates neutral responses and green colors indicate favorable assessments, characterized by specific adjectives mentioned in the figure.

**Figure 9 medicina-59-00617-f009:**
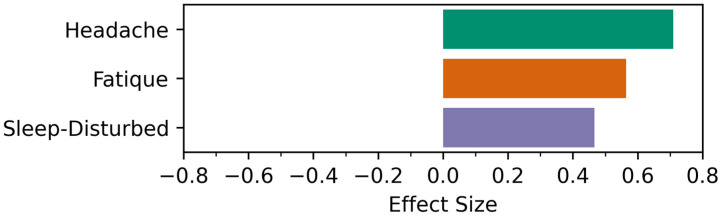
The effect size for the Wilcoxon signed-rank test between the intervention and control period for symptom-frequency-based biomarkers after the first or second vaccination.

**Figure 10 medicina-59-00617-f010:**
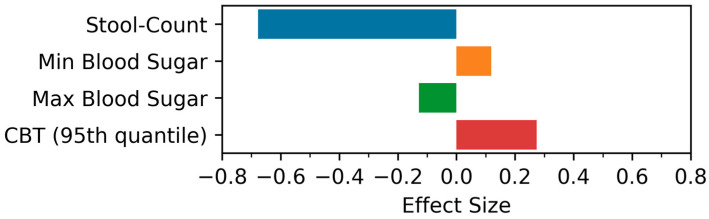
The effect size for the Wilcoxon signed-rank test between the intervention and control period for symptom-intensity-based biomarkers after the first or second vaccination. CBT: Core Body Temperature Sensor.

**Table 1 medicina-59-00617-t001:** Overview of the sensor and adherence thresholds.

Relevant Modalities ^a^	Threshold	Adherence Day Threshold
Physical activity, wrist-worn ^a^	Step count; (threshold) use >20 h/day	Steps ≥ 100
Core Body Temperature ^a^	Continuous Temperature measurements; use >20 h/day	Recording ≥ 1 h
Symptoms, Signs and Measurements ^a^	COVIDA symptom and measurement log questionnaire; at least one answer provided	Completed questionnaire
Spirometry ^a^	Forced expiratory volume in 1 s, forced vital capacity; both values provided	≥3 valid exhalations
Forehead Infrared Temperature skin sensor ^a^	Forehead Temperature: value provided	Recording ≥ 1 h
Nocturnal Pulmometry SpO2-Ring ^b^	Oxygen saturation; use >5 h/day	Recording ≥ 1 audio file
Blood pressure and blood glucose monitoring ^b^	diastolic/systolic value provided, blood glucose result provided	Recording ≥ 1 result

^a^ Measurements were requested from all participants (mandatory measurements). ^b^ Measurements were not considered primary outcome parameters and therefore patients were encouraged to use the device at least once or twice to get to know the device and to be able to assess it in the user experience questionnaire (voluntary measurements).

**Table 2 medicina-59-00617-t002:** Patient baseline characteristics.

Characteristics	Lung Transplant Recipients (*n* = 10)
Age (years), median (range)	47.5 (19–62)
Sex (female/male), n (%)	4/6 (40%/60%)
Underlying diagnosis, no	2 Pulmonary fibrosis 3 COPD 5 Cystic fibrosis
Time since transplant, median (range), days	1579.5 (40–6842)
Forced expiratory volume in 1 s (FEV1) (% predicted), median (range)	77.5 (40–130)
Forced vital capacity (FVC) (% predicted), median (range)	76.5 (36–120)
Smoking status (yes/no), n (%)	No 0/10 (0%/100%)

FEV1, Forced expiratory volume in one second; COPD, chronic obstructive lung disease; FVC, Forced vital capacity.

**Table 3 medicina-59-00617-t003:** Additional device-specific patient comments on different COVIDA desk components.

Comment No	Text Information Provided by Patient	Comment by Research Team
1	It is very exciting for me to see how my vital parameters are and additionally also my daily step count	Positive feedback
2	Monitoring OK for vaccination period, but not for every day all week. Measurements 3× per week would be better/ideal. Especially when used over long periods of time.	Positive feedback
3	The apparatus did not function well, therefore it was not appropriate for me. The controlling/surveillance is too extreme.	Negative feedback
4	The COVIDA app should have a correction button or at least give one the option to go back one step. Otherwise, clear and good.	Suggestion improvement software
5	The app (COVIDA app) unfortunately gets stuck so that one quite often spends more time entering the vital parameters	Technical issue
6	COVIDA app: the symptom log should be better editable	Suggestion improvement software
7	COVIDA app: the entry of data should be possible until 23.59 (not only until 23:00)	Suggestion improvement software
8	The Core thermometer sometimes falls off	Usability issue
9	One has to be careful not to lose the CORE thermometer depending on the trousers one wears.	Usability issue
10	The Core thermometer is interesting. When one is feeling well I would not continue using it 24/7 since one measures the temperature with the forehead infrared thermometer	Addressing double measurements by two methods
11	Suggest a Core thermometer with integrated chest belt analogue to “Polar T31 chest belt”.	Suggested improvement hardware
12	The wrist accelerometer was for me a bit uncomfortable	Adverse device event
13	The accelerometer can lead to wrong statements, for example in court	Not totally clear what the patient means by this.
14	The Ring-Pulse oximeter disturbs when used for longer durations, especially when one is still awake	Usability issue
15	The Ring-Pulse oximeter is not a good idea for sleeping it always falls off.	Usability issue
16	Spirometer: I would prefer an oval mouthpiece	Suggested improvement hardware

The user experience questionnaire contained two questions with a free text field for additional written feedback. All answers provided are compiled in Table 3 and sorted by topics. The number of comments was not limited by space, so some patients made multiple comments.

**Table 4 medicina-59-00617-t004:** Acceptability for continued use of monitoring system for the entire COVIDA desk and specific devices *.

Pat No	COVIDA-Desk (Y or N)	Ring-Pulsoxy	CORE Temp	Accelero-Meter	Infrared Thermometer	COVIDA App	Spirometer
1	Y	Neutral	(Y)	Neutral	Y	Y	Y
2	Y	(N)	(Y)	(Y)	Y	(Y)	Y
3	Y	Neutral	Neutral	Y	Neutral	(Y)	(Y)
4	N	Neutral	(N)	Y	Y	Y	Y
5	Y	Neutral	(Y)	Y	(Y)	(Y)	(Y)
6	N	N	(N)	Y	Y	Y	(Y)
7	N	(N)	(N)	Neutral	(Y)	(N)	(Y)
8	Y	(Y)	Y	Y	(Y)	Y	Y
9	Y	(N)	(N)	Y	Neutral	(Y)	Y
10	N	(N)	N	N	(Y)	(N)	(Y)
Y or (Y)	6	1	4	7	8	8	10
Neutral		3	1	2	2	0	0
N or (N)	4	5	5	1	0	2	0

The question was “Would you be willing to continue monitoring with one or more of the devices?” The answer options included a 5-point scale ranging from “yes”, “partly yes”, “neutral answer”, “partly no” to “no”. Abbreviations: Y = Yes, (Y) = partly Yes, Neutral = Neutral answer, N = No, (N) = partly No. All questions had 5 possible levels of answer, except the first question where only “yes” or “no” answers were allowed.

## Data Availability

Data available upon reasonable request.

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
