# Peer review of "Multimodal Remote Home Monitoring of Lung Transplant Recipients during COVID-19 Vaccinations: Usability Pilot Study of the COVIDA Desk Incorporating Wearable Devices"

_medicina, 2023, doi:10.3390/medicina59030617_

Round 1
Reviewer 1 Report
This article presents a study for monitoring lung transplant recipients during Covid vaccination. The system monitors the effects of vaccinations on these patients.
The work is well presented and well framed in terms of related work and state of the art.
Also in the discussion the results obtained are framed and compared with other publications.
Overall the study is solid, however the number of patients is only 10 which is very little, however, it may be enough to minimally validate the results obtained. The age range of the patients is also very wide, certainly, a 19-year-old patient has a different response than a 66-year-old patient. I don't even know if it wouldn't have been more prudent to minimize the age gap.
However, the work is valid.
Author Response
Point-by-point response to the reviewer comments:
Reviewer1:
This article presents a study for monitoring lung transplant recipients during Covid vaccination. The system monitors the effects of vaccinations on these patients.
The work is well presented and well framed in terms of related work and state of the art.
Also in the discussion the results obtained are framed and compared with other publications.
Answer: Thank you for these comments. No action is needed concerning these points.
Overall the study is solid, however the number of patients is only 10 which is very little, however, it may be enough to minimally validate the results obtained. The age range of the patients is also very wide, certainly, a 19-year-old patient has a different response than a 66-year-old patient. I don't even know if it wouldn't have been more prudent to minimize the age gap.
Answer: Your points are valid and well-taken. The number of patients is small, and the age range is very wide. In this respect, the repeated review of the data showed a transcription error in the birth year of the oldest patient, who is actually62 years (and not 66, as had been reported). Nevertheless, the age range remains very wide, which we now mention in the Discussion section. It was not our aim to limit the age range and we were interested in studying the monitoring devices in different age groups due to suspected variability in the acceptance. Surprisingly, the acceptance of the COVIDA desk was good among the studied patients, despite the broad age group. Based on the comments from both reviewers, we tested for an association between age or time since transplantation and the adherence. We did not find a significant association. We have added these findings in the Results section (lines 272-273 in CLEAN version of manuscript) and mentioned them in the Discussion section (lines 428-432; lines 516-517).
However, the work is valid.
Answer: Thank you for your comments and observations.
Reviewer 2 Report
Dear authors,
It is a pleasure for me to review your manuscript.
Remote patient monitoring is a spreading strategy to care for patients while limiting the effect of medical controls on patients’ quality of life.
Lung transplant recipients are fragile patients who require to be monitored, especially in such new conditions like during vaccine administration.
I found your paper a well conducted pilot study to test the COVIDA Desk; the strengths and the limitations of the device as well as user cpmpliance have been well presented.
From my point of view, your text requires minor revisions:
Concerning the cohort of subjects selected, patient baseline characteristics are presented in Table 2. Both age and time since transplant have a large variability. Did you consider if the age and the time from the transplant affect the adherence of the subject in Covida Desk use? I understand that the sample is small, but I would really appreciate if you could put this information in the text. Age and time since transplant can give relevant details about the compliance of the subjects in using remote patient monitoring.
Please verify in the text, on line 7 there is an inappropriate use of the word “and”. On the line 40 some keywords are reported in bold, and "usability; feasibility; adherence" do not seem to me appropriate keywords, while for example COVID-19 vaccination and lung transplant could fit better.
Figures 6 and 7 are not presented in the text. Please consider to remove or to describe their results in the text.
Author Response
Point-by-point response to the reviewer comments:
Reviewer 2:
Dear authors,
It is a pleasure for me to review your manuscript.
Remote patient monitoring is a spreading strategy to care for patients while limiting the effect of medical controls on patients’ quality of life.
Lung transplant recipients are fragile patients who require to be monitored, especially in such new conditions like during vaccine administration.
I found your paper a well conducted pilot study to test the COVIDA Desk; the strengths and the limitations of the device as well as user compliance have been well presented.
Answer: Thank you for these comments, which do not require modifications to the manuscript.
From my point of view, your text requires minor revisions:
Concerning the cohort of subjects selected, patient baseline characteristics are presented in Table 2. Both age and time since transplant have a large variability. Did you consider if the age and the time from the transplant affect the adherence of the subject in Covida Desk use? I understand that the sample is small, but I would really appreciate if you could put this information in the text. Age and time since transplant can give relevant details about the compliance of the subjects in using remote patient monitoring.
Answer: Thank you for this insightful remark and the question concerning influence of age and time since transplantation (both with large variability) on the adherence to the COVIDA desk use.
Based on your suggestion, we performed additional analyses to detect an association between the age and time from transplantion and adherence.
To assess these relationships, we calculated median adherence across all modalities for each patient resulting in a single adherence statistic.
- We split patients based on the median age (47.5) into young (median adherence for this group = 88%) and elderly (median adherence = 81%) group. Then, we compared median adherence across modalities for those two groups by the means of the two-sided Wilcoxon rank sum test (statistic: 0.0, p-value: 1.0, effect size 0.0). Conclusion: There is no association between the patient age and adherence.
- We split patients based on the median time since transplant (1579.5 days) into two groups: short period (median adherence for this group = 94%) and long period (median adherence = 75%). Then, we compared the median adherence across modalities for those two groups by the means of the two-sided Wilcoxon rank sum test (statistic: -0.94001934, p-value: 0.34720764, effect size: -0.29726022). -0.29726022 is a small negative effect size. That means that patients who have recently had transplantation tended to adhere more than the other patients. However, it is not statistically significant. Conclusion: There is no significant association between the time since transplant and adherence.
We included this information in the Methods section (lines 230-233, lines 247-249) and in the Results section (lines 272-273) without overemphasizing its importance, due to the small sample size. Additionally we discuss it in the Discussion section (lines 428-432; lines 516-518).
Please verify in the text, on line 7 there is an inappropriate use of the word “and”. On the line 40 some keywords are reported in bold, and "usability; feasibility; adherence" do not seem to me appropriate keywords, while for example COVID-19 vaccination and lung transplant could fit better.
Answer: Thank you for these suggestions, which we have implemented in the manuscript accordingly (lines 300-302).
Figures 6 and 7 are not presented in the text. Please consider to remove or to describe their results in the text.
Answer: Thank you for this relevant comment. We have described the findings of Figures 6 and 7 in the manuscript’s text.
We would like to like to thank both reviewers for their constructive comments and suggestions, which have helped improve the manuscript’s message and clarity.
Macé Schuurmans & Co-authors, 13.3.2023
